# Does the Administration of Intravenous Fluid Matter in the Context of the Incidence of Postoperative Complications After Radical Cystectomy?

**DOI:** 10.3390/cancers17010102

**Published:** 2024-12-31

**Authors:** Paweł Lipowski, Adam Ostrowski, Jan Adamowicz, Przemysław Jasiewicz, Filip Kowalski, Tomasz Drewa, Kajetan Juszczak

**Affiliations:** 1Department of Urology and Andrology, Collegium Medicum, Nicolaus Copernicus University, M. Curie Skłodowskiej 9, 85-094 Bydgoszcz, Poland; adostro@gmail.com (A.O.); adamowicz.jz@gmail.com (J.A.); urologpolska@gmail.com (F.K.); t.drewa@wp.pl (T.D.); 2Department of Regenerative Medicine, Collegium Medicum, Nicolaus Copernicus University, M. Curie Skłodowskiej 9, 85-094 Bydgoszcz, Poland; 3Department of Anesthesiology and Intensive Care, Collegium Medicum, Nicolaus Copernicus University, 85-094 Bydgoszcz, Poland; pjasiewicz@cm.umk.pl

**Keywords:** radical cystectomy, bladder cancer, ERAS protocol, fluid therapy

## Abstract

Intravenous fluid management plays a key role in perioperative care, particularly in patients undergoing radical cystectomy (RC) within enhanced recovery after surgery (ERAS) protocols. This study analyzed 288 patients who underwent laparoscopic RC with urinary diversion to evaluate the impact of fluid administration on postoperative complications assessed 30 days after surgery. Patients were categorized based on the type of urinary diversion (ureterocutaneostomy or ileal conduit) and the volume of intraoperative fluids administered (less than or more than 1000 mL). The results showed that administering more than 1000 mL of fluids was initially associated with an increased risk of complications, but this association lost statistical significance after adjusting for surgery duration and BMI. Indices such as the absolute Vascular Bed Filling Index (aVBFI) and the adjusted Vascular Bed Filling Index (adjVFBI) revealed differences in complication severity 30 days after surgery depending on the type of urinary diversion and fluid management strategy.

## 1. Introduction

Intravenous fluid restoration is an essential part of perioperative and postoperative care. Adequate intravenous fluid administration maintains proper water and electrolyte balance. The optimal amount of intravenous fluids during and after general surgery still remains undefined [1]. Previous studies have revealed that personalized perioperative fluid management has improved the duration of postoperative ileus (with a lower complication rate) [1,2]. According to some surgical guidelines, fluid intake should be restricted to a maximum of 3 L on the day of surgery [3].

It is worth noting that a restricted postoperative fluid regime reduces complications and postoperative hospital stay after surgery, especially when using the ERAS (Enhanced Recovery After Surgery) protocol [4,5]. Upon achieving promising results in general surgery, the ERAS protocol also began to be implemented in patients undergoing urological procedures. Radical cystectomy (RC) remains the gold standard for treating muscle-invasive urinary bladder cancer, but it is associated with a significant risk of complications and mortality [6,7,8]. It is worth noting that LND is associated with higher perioperative morbidity and mortality in patients with muscle-invasive bladder cancer undergoing radical cystectomy. Moreover, extended LND did not result in improved disease-free or overall survival in comparison with standard LND [9]. Moreover, Pfail et al. [10] showed that patients with muscle-invasive bladder cancer and with clinically positive lymph nodes, especially in N2-N3 disease, require a tailored approach. While neoadjuvant chemotherapy followed by radical cystectomy is standard for N1 disease, the role of surgery in advanced nodal stages is growing because of better patient selection and treatment strategies [10]. Neoadjuvant chemotherapy improves cancer-specific mortality (CSM) compared to RC alone, both in organ- and non-organ-confined urinary bladder cancer [11]. Additionally, de Angelis et al.’s [12] study showed that neoadjuvant chemotherapy before RC does not result in a statistically significant CSM benefit relative to RC followed by adjuvant chemotherapy.

The implementation of the ERAS protocol for patients undergoing RC can reduce the hospitalization time and may improve the cancer-specific survival (CCS) and the 5-year overall survival (OS) [13,14,15,16]. Moreover, meta-analyses demonstrate that applying the ERAS protocol in patients undergoing RC reduces the complication rate and accelerates the return of bowel function [17]. The introduction of less invasive modern surgical techniques allows for a reduction in hospitalization time and a decrease in the number of complications [18].

While working on the study, we hypothesized that excessive fluid administration during cystectomy could lead to edema of the intestinal-to-intestinal or uretero-intestinal anastomoses, particularly in patients from the ileal conduit group. Additionally, the ERAS protocol and its components are implemented differently across various centers. With this study, we aimed to emphasize the importance of restrictive fluid management and the necessity of considering vascular bed filling.

From a pathophysiological point of view, proper water, electrolyte, protein and acid–base balance is crucial for the quick recovery of patients undergoing major surgery. Thus, restrictive fluid therapy remains an important component of the ERAS protocol.

Currently, knowledge regarding the influence of fluid supply during RC is scarce. Several studies on the optimal method of fluid therapy in patients undergoing RC have been conducted by both anesthesiologists and urologists. Previous randomized clinical trials evaluating various fluid therapy regimens did not conclusively determine the management of fluid therapy in patients undergoing RC for urinary bladder cancer [19,20,21,22].

Previous knowledge regarding the effect of fluid therapy during and after RC is still inconclusive.

Therefore, the main purpose of this study was to investigate the effects of intravenous fluid administration on postoperative complications in patients undergoing RC.

## 2. Materials and Methods

### 2.1. Study Protocol

Two hundred eighty-eight consecutive patients with muscle-invasive urinary bladder cancer who underwent laparoscopic RC with lymphadenectomy and urinary diversion (ureterocutaneostomy or ileal conduit) from 2018 to 2022 were enrolled in a retrospective cohort study. The same surgeon performed all RCs. In each patient of our cohort who underwent radical cystectomy, lymph node dissection (LND) was performed. The standard LND in muscle-invasive bladder cancer involves the removal of nodal tissue cranially up to the common iliac bifurcation, with the ureter being the medial border, and includes the internal iliac, presacral, obturator fossa and external iliac nodes. The lateral borders are the genitofemoral nerves, caudally the circumflex iliac vein, the lacunar ligament and the lymph node of Cloquet [11]. The cohort included patients of various cancer stages, ages and comorbidities.

Patients after RC were divided into four groups based on the urinary diversion technique [ureterocutaneostomy (UCS) and ileal conduit (IC)] and the amount of intravenous fluid (crystalloids) administration [< or >1000 mL], as follows: (1) group 1 [UCS + fluid intake < 1000 mL], (2) group 2 [UCS + fluid intake > 1000 mL], (3) group 3 [IC + fluid intake < 1000 mL] and (4) group 4 [IC + fluid intake > 1000 mL].

ERAS protocols were implemented in each patient who underwent RC, incorporating both preoperative and postoperative measures.

Preoperative measures included promoting increased physical activity and a high-protein diet approximately four weeks prior to RC. No bowel preparation or enemas were used. Carbohydrate- and electrolyte-rich fluids were administered the day before and the morning of the surgery to mitigate insulin resistance. Perioperatively, gabapentin was employed as a co-analgesic. Postoperative management allowed fluid intake on the day of surgery, with oral nutrition initiated on the first postoperative day. Gastric tubes were not used postoperatively, and the route of drug administration was switched to the oral route as tolerated. Low-molecular-weight heparin was administered for thromboprophylaxis.

The collected data included gender, age, body mass index (BMI), Charlson comorbidity index (CCI), nutrition scale (nutrition risk screening, NRS), anesthesiological risk using the ASA (American Society of Anesthesiologists) score, baseline hemoglobin levels and the presence of comorbidities such as hypertension, type 2 diabetes, heart failure, where patients with heart failure were classified as those diagnosed with a reduced left ventricular ejection fraction (LVEF) of <50% during cardiological evaluation, and chronic kidney disease.

Several parameters closely related to the surgical procedure, such as the operation time, blood loss and the method of urinary diversion (UCS or IC) were also taken into account.

These groups were compared in terms of complications at 30 and 90 days post-surgery, and 30- and 90-day post-surgery complications and their relationships were assessed in all experimental groups. Complications were assessed using the Clavien-Dindo scale.

In assessing complications, the ratio of the volume of fluids administered relative to body weight (mL/kg), BMI (mL/kg/m^2^), duration of the procedure and combined ratio, taking into account procedure time and body weight (mL/hour/kg), were used. Additionally, two novel clinical parameters were introduced (see Section 2.2).

Propensity score matching (PSM) was conducted to balance baseline characteristics between the UCS group (*n* = 240) and the ileal conduit diversion group (n = 48). To achieve covariate balance between the groups, the validity of subsequent comparisons was enhanced. PSM was implemented using a nearest-neighbor matching algorithm with a 1:1 matching ratio, ensuring that each patient in the IC group was matched with one patient from the UCS group based on similar propensity scores. The propensity scores were estimated using a logistic regression model, with patient age, CCI (Charlson Comorbidity Index) and clinical tumor stage as predictors.

### 2.2. Novel Clinical Parameters

Due to the high heterogeneity of patients undergoing RC related to different patient weights and operation time (which is influenced, among others, by the type of urinary diversion), we have defined two novel clinical parameters (aVBFI and adjVFBI), which allow for the unification of experimental groups, allowing for a reliable comparison between groups.

The absolute Vascular Bed Filling Index (aVBFI) assesses vascular bed filling during the procedure. The aVBFI is calculated as the ratio of fluids administered during the procedure due to blood loss.

The adjusted Vascular Bed Filling Index (adjVFBI) is a more accurate parameter than aVBFI because it also considers the procedure’s duration. The adjVFBI is calculated as the ratio of the aVBFI to the duration of the procedure in minutes.

Moreover, in further analysis, we assumed that fluid loss through evaporation, respiration and convection remains constant, and that in laparoscopic interventions, it is not as significant as in open surgeries.

### 2.3. Statistical Analysis

Data analysis was conducted, adhering to rigorous methodological and statistical standards, with a significance level set at α = 0.05, which is standard in biomedical sciences, minimizing the risk of Type I error. Descriptive statistics tailored to the type of data were utilized. For quantitative variables, the median was used as it is less sensitive to outliers than the arithmetic mean, and the first and third quartiles were presented, allowing for a fuller understanding of the data distribution.

For categorical variables, the number of observations and percentages were used, facilitating the interpretation of the distribution. To assess the significance of differences between groups, the Wilcoxon rank-sum test was used for quantitative variables and the Pearson Chi-square test, Fisher’s exact test or a proportions test for categorical variables, depending on the sample size and data characteristics.

A multivariate analysis was conducted using a multiple regression model to identify factors affecting the severity of postoperative complications, enabling the assessment of relationships between multiple independent and dependent variables. A robust estimator (M-Huber) that is less sensitive to outliers was used in the analysis, ensuring greater reliability of the results.

The analysis provides information on the risk profile and enables the identification of high-risk groups as well as risk factors that can be targeted for intervention to improve treatment outcomes. The statistical analysis results are presented with 95% confidence intervals and *p*-values, allowing for the precise interpretation of the strength and direction of the statistical relationships.

Propensity score matching analyses were conducted using the R Statistical language (version 4.3.3; R Core Team, 2024) on Windows 11 pro 64 bit (build 22631), using the packages cobalt (version 4.5.5; Greifer N, 2024), MatchIt (version 4.5.5) [23].

## 3. Results

### 3.1. Characteristics of Experimental Groups

The median age of the patients was 68.5 years (63 to 74 years). The gender distribution in the study group showed a significant predominance of males, who comprised 78.1% (225 patients), compared to females, who accounted for 21.9% (63 patients). The median body weight of the patients was 80 kg (70 to 89 kg). The median BMI was 26.95 kg/m^2^ (24.30 to 30.05 kg/m^2^). Ureterocutaneostomy and ileal conduit were conducted in 240 and 48 patients, respectively.

The hospitalization stay was 10 days (8 to 13 days). The duration of hospitalization did not differ significantly between all experimental groups (*p* = 0.153).

Hypertension (HT) was the most common comorbidity, which affected 65.3% of all patients. In patients with IC, the incidence of HT was lower at 54.1% compared to the group with UCS, which had a prevalence of 67.5%. However, this difference did not reach statistical significance (*p* = 0.077). Diabetes mellitus (DM) was present in 24.7% of the participants, with a slightly higher frequency in the patients with IC (29.2%) compared to the patients with UCS (23.8%), although this difference was also not statistically significant (*p* = 0.427). Similarly, the incidence of heart failure (HF) did not differ significantly between groups, with 14.6% in the IC and 19.1% in the UCS patients (*p* = 0.454). In our cohort, a statistically significant difference was observed in the case of chronic kidney disease (CKD). In the group of patients after IC, the percentage of patients with CKD was significantly lower (8.3%) compared to the patients with UCS (30%), with a *p*-value of 0.002. (Table 1)

Patients in the IC group were statistically younger than those in the USC group. The median age for the IC group was 64 years, compared to 69 years for the USC group (*p* < 0.001).

According to the Charlson comorbidity index (CCI) results, the majority of patients (59.7%) had a score greater than four, suggesting the presence of comorbid conditions that worsen the prognosis. A statistically significant difference (*p* < 0.001) between urinary diversion techniques indicates that a higher percentage of patients with ileal conduit had a better CCI outcome (<4) (64.6%) compared to the patients with UCS (35.4%).

For older and comorbid patients, the uretero-ureterocutaneous fistula (UCN) without the use of intestinal interposition devices is an alternative and safe procedure [24]. In our population of 417 patients, the operation time (incision–suture time: median 82 min; range 63–121 min) and mortality (2.1%) are low despite a pronounced negative selection of comorbid in elderly patients (mean age 79 years). A potential disadvantage compared to the ileum conduit is the obligatory ureteral splinting.

The assessment of the ASA scale score revealed significant differences between the groups (*p* = 0.004). In the patients with IC, a larger percentage of patients had a lower risk class, ASA2 (41.7%), compared to the UCS patients (19.2%), while the higher risk class, ASA3, was more common in the UCS patients (79.2%) than in the IC patients (58.3%). The NRS (nutritional risk score) also showed a statistically significant difference (*p* = 0.022), with the median score in the IC patients being lower than that in the UCS patients.

The study observed a statistically significant difference in preoperative hemoglobin (HGB) levels between the groups, where patients with IC creation had a higher median HGB (13.80 g/dL) compared to the patients with UCS (12.30 g/dL). The differences remained significant postoperatively, with higher HGB in the IC patients (11.95 g/dL) compared to the UCS patients (10.65 g/dL). The loss of hemoglobin did not differ significantly between the groups.

The use of neoadjuvant chemotherapy (CHTx) in this cohort was relatively low, at 15.3% (44 individuals)—5 patients from the IC group (10.4%) and 39 patients from the UCS group (16.3%). The neoadjuvant CHTx used for our patients was based on cisplatin agents.

The low percentage of patients receiving neoadjuvant chemotherapy results from the significant comorbidity of the Polish population compared to other European populations [25]. Scientific reports indicate that the risk of malnutrition in the Polish population over 65 years old can reach 44.2% [26] and 39.2% in the oncological patient group [27]. The prevalence of overweight is 63% in men and 43% in women [28]. This also explains why the simplest form of urinary diversion is often chosen after cystectomy (CR).

The results indicate statistically significant differences (*p* < 0.05) in terms of clinical tumor stage (cT). In the UCS group, individuals with advanced cT4 tumors constituted 27.5%, while the patients with IC constituted only 4.17% (*p* = 0.001). (Table 1) A higher proportion of patients in the UCS group was suspected of having lymph node metastases compared to the IC group (13.3% vs. 2.1%, *p* = 0.025). Patients in the IC group more frequently had organ-confined disease (cT2)—52.1%, compared to 31.7% in the UCS group (*p* = 0.007). The percentage of patients with cT3 advancement did not differ significantly between the groups, with 41.6% in the IC group and 40.1% in the UCS group (*p* = 0.915). However, the presence of hydronephrosis was significantly less common in the IC group (10.4%) compared to the UCS group (34.2%) [*p* = 0.001].

Preventing infections, including wound infections, appears to play a crucial role during radical cystectomy and in the perioperative period. Infectious complications can account for up to 25% of complications following radical cystectomy procedures during RARC [29]. If open access is used during radical cystectomy (ORC), the use of Alexis wound protectors has been shown to effectively reduce the rate of postoperative wound infections [30]. In our study, all procedures were performed laparoscopically, and we used standard extraction bags. We observed only incidental cases of wound infections or pneumonia after laparoscopy.

The median operation time was 200 min (from 175 to 241 min.). The duration of the procedure was significantly longer in the group of patients using the IC creation (average 260 min.) compared to the UCS creation (average 190 min.; *p* < 0.001). The median blood loss was 250 mL (ranging from 150 to 350 mL). Concurrently, there was no significant difference in blood loss between the UCS and IC groups (*p* = 0.962). (Table 2).

The ratio of fluids administered per hour of surgery per kilogram of body weight (mL/hour/kg) was significantly lower in the IC group (3.69) compared to the UCS group (4.41; *p* < 0.001).

This may be due to the fact that patients in the IC group statistically weighed more than those in the USC group—84 kg vs. 79 kg (*p* = 0.011). However, no differences between the groups were noted after calculating the BMI.

The rate of blood product transfusions was higher in the UCS group (34.2%) than in the IC group (14.6%; *p* = 0.007). No statistically significant differences were observed in the percentage of patients requiring intensive care unit support (*p* = 0.228).

There was a statistically significant difference in the incidence of ileus (*p* = 0.001), with a higher frequency of cases in the IC group (27.1%) compared to the UCS group (9.6%).

### 3.2. Postoperative Complications According to the Clavien-Dindo Classification Within 30 Days

The frequency of complications, as measured by the Clavien-Dindo scale, did not differ significantly, regardless of the method of urinary diversion or the amount of fluids administered. The distribution of complications in each group is attached in Table 3.

However, several factors that influence the severity of complications were identified (Table 4).

The administration of fluid volumes above 1000 mL is associated with a higher severity of complications (*p* = 0.035) within 30 days post-surgery when not accounting for additional factors such as the duration of the operation and patient weight. However, the lack of statistical significance in this area after adjusting the volume of administered fluids for the duration of the operation and patient BMI indicates that fluid volume alone is not a reliable predictor of the severity of complications after RC. Using the UCS after RC is associated with less severe complications according to the Clavien-Dindo scale, suggesting its potential benefits in a postoperative context (B = −0.53, *p* = 0.024). However, the significance of this variable becomes more complex when considering its interaction with the absolute Vascular Bed Filling Index (aVBFI). A direct comparison of the four experimental groups, based solely on the number of fluids administered (< or >1000 mL), showed no differences.

However, after applying the aVBFI, it was found that there was a significant interaction between aVBFI and the type of urinary diversion (UCS/IC) (B = 0.07, *p* = 0.011).

At aVBFI values between zero and eight, urinary diversion using the UCS is associated with a lower risk of complications compared to the IC. When aVBFI equals eight, the differences in the severity of complications between the UCS and the IC are minimal. However, when aVBFI exceeds eight, the IC creation is associated with fewer complications within 30 days postoperatively compared to UCS (Figure 1).

The correlation between the adjusted Vascular Bed Filling Index (adjVFBI) (B = −0.27; 95% CI: −0.45 to −0.08; *p* = 0.005) and the severity of complications up to 30 days postoperatively is similar to that seen with aVBFI. Similarly, the correlation of adjVFBI with the method of urinary diversion (B = 0.24; 95% CI: 0.06 to 0.43; *p* = 0.011) resembles that of aVBFI.

### 3.3. Complications According to the Clavien-Dindo Classification After 90 Days

The different amounts of intravenous fluid administration did not influence the severity of complications in 90 days postoperatively.

The statistical analysis based on two hundred and eighty observations (eight patients died before reaching 90 days post-surgery) indicated that the occurrence of acute kidney injury in the postoperative period was associated with a higher degree of complication severity (B = 0.40; 95% CI: 0.04–0.75; *p* = 0.028). A longer hospitalization time was also significantly associated with a higher degree of complication severity (B = 0.04; 95% CI: 0.02–0.06; *p* < 0.001). No significant associations were observed between other factors and complications after 90 days from the procedure.

### 3.4. Propensity Score Matching

The propensity score matching procedure was successful in balancing covariates between the UCS group and the ileal conduit diversion group. Prior to matching, significant imbalances were observed across several covariates. For instance, the standardized mean differences (SMD) indicated substantial differences in age (SMD = −0.80), CCI scale (“good” vs. “burdened” SMD = 0.61 and −0.61, respectively) and tumor stage, particularly for cT4 (SMD = −1.17). These imbalances suggest that patients in the UCS group were older, had a higher burden of comorbidities and were more likely to present with advanced tumor stages compared to the ileal conduit group.

After matching, covariate balance improved considerably. The standardized mean differences for all covariates were reduced to within acceptable thresholds, with SMD values for age (−0.08), CCI categories (0.00 for both (“good” vs. “burdened”) and tumor stages (e.g., cT1-cT2: SMD = 0.13, cT3: SMD = −0.17, cT4: SMD = 0.10) indicating minimal residual imbalance. The mean difference in propensity scores between the groups was also reduced (SMD = 0.05), reflecting the successful alignment of treatment probabilities between the matched groups. The variance ratios for continuous variables, such as age (0.72), and eCDF differences for categorical variables (all ≤0.13) further confirm improved balance post-matching (see Figure 2 for visualized covariate balance).

The matching process retained all 48 patients in the ileal conduit group and matched them to 48 patients in the UCS group, reducing the control group from 240 to 48 for direct comparisons. This ratio ensured equal representation of both groups in the matched dataset, minimizing bias in subsequent analyses. Importantly, no patients were discarded or excluded due to extreme propensity scores, indicating that overlap in treatment probabilities was sufficient for reliable matching.

These results imply that the matched dataset provides a robust basis for comparing outcomes between the UCS and IC groups, free from the confounding influences of age, comorbidities and tumor stage. The improved balance after matching allows for more accurate estimation of the effect of urinary diversion type on patient outcomes, without the bias introduced by baseline differences between the groups.

The assessment of the impact of the urinary diversion method using the covariate balance sample (N = 96, n IC = n UCS = 48) reproduced the results of the previously reported regression models. The UCS diversion was associated with a significantly lower severity of complications one month after surgery (B = −0.51, 95% CI: −0.99–−0.03, *p* = 0.036) compared to the IC diversion. However, the urinary diversion method did not have a significant effect on the Clavien-Dindo parameter at three months post-surgery (*p* = 0.415).

## 4. Discussion

The results of this study indicate that the amount of intravenous fluids administration as an isolated parameter does not correlate with the severity of early and late postoperative complications after RC. The significant effects observed with newly defined clinical parameters such as absolute and adjusted Vascular Bet Filling Index (aVBFI and adjVBFI) suggest that the appropriate ratios of fluid resuscitation relative to blood loss are crucial for improving postoperative outcomes and can effectively reduce the severity of complications. Our findings also suggest that simpler methods of urinary diversion should be considered during procedures associated with significant blood loss.

In various single-center studies of radical cystectomy, the incidence of complications remains high even when tumors are slightly less advanced. Additionally, the rate of blood transfusions has been reported to be as high as 54.8%, as detailed in the study by Anil Erdik et al. [31].

Bazargani et al. [19] found no correlation between the amount of perioperative fluids input (including crystalloids, colloids, and blood products) and complications at 30 and 90 days post-surgery in 180 patients after open RC. Additionally, fluid administration did not affect the length of hospital stay.

On the other hand, Wuetrich et al. [32] randomized 166 patients into two groups and demonstrated that reduced fluid administration at a rate of 3 mL/kg/hour during surgery, combined with intraoperative noradrenaline infusion, reduced complications by 21% (52% vs. 73%) compared to administering 6 mL/kg/hour without noradrenaline. This approach also decreased the risk of death within 90 days after RC by 4.8% (0% vs. 4.8%). However, their study did not utilize the ERAS protocol, whose components may positively influence the frequency of complications and postoperative stress. This might be the reason for conflicting results regarding fluid therapy alone in these patients.

Similarly, Furrer et al.’s [33] study, in a group of 775 patients undergoing RC without the ERAS protocol, demonstrated that a reduced crystalloid administration of 3.5 vs. 4.2 mL/kg body weight during the procedure may contribute to developing postoperative renal failure. However, no effect of colloids on postoperative renal failure was demonstrated.

Studies emphasize that fluid administration should be individualized and based on objective measurements of flow rates to achieve optimal preload values. This involves adjusting fluid delivery according to the filling of the vascular bed, which can be monitored using tools like an esophageal Doppler probe placed in the patient. Goal-directed fluid therapy (GDFT) has been particularly studied in terms of the recovery of gastrointestinal (GI) function after surgical procedures, and several meta-analyses have been conducted on this topic.

Giglio and colleagues analyzed 16 studies and demonstrated that GDFT reduces the frequency of severe GI complications compared to the control groups [34]. Additionally, Gomez-Isquierdo et al. [1] concluded that GDFT decreases the incidence of postoperative nausea and vomiting, allows patients to tolerate a diet sooner and shortens the time to the first bowel movement. These benefits did not extend to patients who were managed under the ERAS protocol. Also, Srinivasa et al. [35] analyzed 691 patients where GDFT was combined with the ERAS protocol in colorectal surgery and found no differences in complications or length of hospital stay among the patients.

The impact of GDFT has also been studied in patients with urinary bladder cancer undergoing RC. After analyzing two fairly large groups of patients who underwent open RC, where GDFT was applied in the experimental group without the ERAS protocol, Arslan-Carlon et al. [22] did not demonstrate a positive impact of this fluid therapy approach on either intestinal function or the number of high-grade postoperative complications. They observed a higher risk of acute kidney injury (AKI) in patients from the GDFT group. On the other hand, Pillai and colleagues demonstrated that using GDFT in patients undergoing RC reduces GI complications and improves wound healing [2]. However, the experimental and control groups were small; the procedure duration significantly differed between groups, and the ERAS protocol was not implemented in this study.

Ghoreifi et al. [20] studied the application of GDFT in patients undergoing RC and the ERAS protocol implementation. They evaluated kidney function, the length of stay, complications and rehospitalization 90 days post-surgery. No significant differences were found between the experimental and control groups. Additionally, stroke volume variation (SVV) was studied by Kong et al. [21] in patients undergoing RC. SVV is gaining increasing attention as a reliable index for protocolized fluid management (specifically maintaining SVV within the 10–20% range) to minimize vascular congestion in surgical areas and intraoperative blood loss during procedures such as living-donor hepatectomy. However, this study did not show the impact of this method of fluid transfusion on postoperative outcomes. Nevertheless, it did help to reduce blood loss and the number of transfusions of blood products.

The results of the studies presented above are conflicting, yet they suggest that intraoperative fluid administration based on the filling of the vascular bed or blood loss may influence the complications following RC.

We would also like to mention that a limitation of our study is the retrospective nature of the data analysis. We also did not exclude patients with advanced cancer or those undergoing palliative cystectomy from the analysis. The group of patients in whom urinary diversion was performed using the IC is significantly smaller, and these patients had lower CCI scores and initial cancer staging. Patients in the UCS group also had a higher risk of malnutrition, which is an independent risk factor for the occurrence of complications 30 days after the procedure [36]. However, comparing the results of our study with those of other researchers suggests that optimizing fluid administration based on the filling of the vascular bed could have a positive impact on complications in patients undergoing RC together with the application of the ERAS protocol.

Additionally, it seems that newly defined clinical parameters, such as absolute and adjusted Vascular Bed Filling Index (aVBFI and adjVBFI), seem useful in everyday urological practice in assessing the risk of postoperative complications after RC.

## 5. Conclusions

The volume of fluids administered does not directly affect the severity of complications following RC when the ERAS protocol is used. The amount of intraoperative fluid administration should be adjusted according to the intraoperative blood loss. In procedures with significant blood loss, simpler methods of urinary diversion should be considered.

The findings endorse the utility of novel clinical parameters such as aVBFI and adjVBFI as valuable clinical tools in guiding fluid therapy within the framework of ERAS protocols. However, further multicenter randomized trials are needed to definitively determine the best fluid therapy regimen for patients undergoing RC.

## Figures and Tables

**Figure 1 cancers-17-00102-f001:**
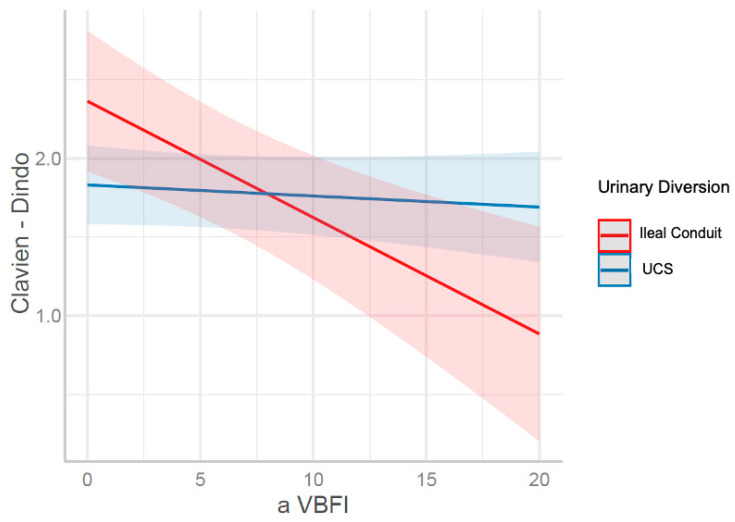
The interaction effect between aVBFI and the urinary diversion on the severity of complications using the Clavien-Dindo scale one month after RC.

**Figure 2 cancers-17-00102-f002:**
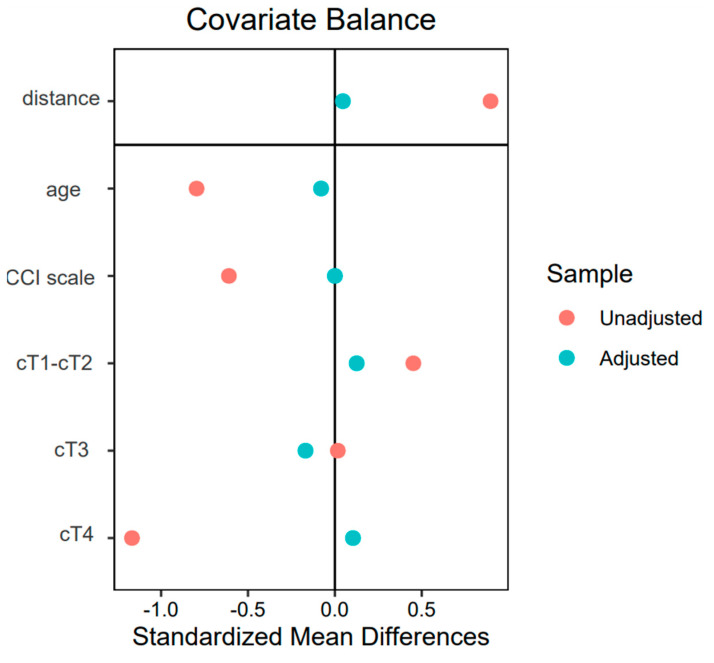
Standardized mean differences for covariate balance before and after propensity score matching.

**Table 1 cancers-17-00102-t001:** Characteristics of the study group.

Characteristic	N = 288	Overall in Sample ^a^	Urinary Diversion		*p* ^b^
			IC ^a^,	UCS ^a^,	
			*n* = 48	*n* = 240	
Hypertension		188 (65.28%)	26 (54.2%)	162 (67.5%)	0.077
Diabetes mellitus		71(24.65%)	14(29.2%)	57 (23.8%)	0.427
Heart failure		53(18.40%)	7 (14.6%)	46 (19.2%)	0.454
Chronic kidney disease		76(26.39%)	4 (8.3%)	72 (30.0%)	0.002
Neoadjuvant chemotherapy		44 (15.3%)	5 (10.4%)	39 (16.3%)	0.305 ^b^
cTNM:					
cT2		101 (35.1%)	25 (52.1%)	76 (31.7%)	0.007 ^d^
cT3		118 (41.0%)	20 (41.7%)	98 (40.8%)	0.915 ^d^
cT4		68 (23.6%)	2 (4.2%)	66 (27.5%)	0.001 ^c^
N1		33 (11.5%)	1 (2.1%)	32 (13.3%)	0.025 ^c^

^a^ n (%); ^b^ Pearson Chi-square test; ^c^ Fisher’s exact test; ^d^ proportions test.

**Table 2 cancers-17-00102-t002:** Parameters of perioperative factors in the sample, stratified by urinary diversion technique after cystectomy.

Characteristic	N = 288	Overall in Sample ^a^	Urinary Diversion		*p* ^c^
			IC ^a^, n = 48	UCS ^a^, n = 240	
Surgery time min		200 (175–241)	260 (242–285)	190 (165–220)	<0.001
Blood loss mL		250 (150–350)	250 (200–350)	250 (150–350)	0.962
Fluids administrated during surgery mL		1000 (500–1900)	1350 (800–1900)	1000 (500–1900)	0.011
Fluids administrated/Surgery time/Weight mL/hr/kg		4.28 (3.34–5.41)	3.69 (2.55–4.60)	4.41 (3.58–5.79)	<0.001
Fluids/BMI ml/kg/m^2^		140.5 (124.0–158.0)	143.0 (124.0–152.25)	140.0 (124.0–159.25)	0.998
aVBFI (1)		4.80 (3.18–7.04)	5.00 (3.63–8.08)	4.33 (3.0–7.0)	0.107

^a^ Mdn (Q1, Q3); ^c^ Wilcoxon rank-sum test.

**Table 3 cancers-17-00102-t003:** Complications depending on the type of urinary diversion type.

Characteristic	N	Overall in Sample ^a^	Urinary Diversion		*p* ^b^
			IC ^a^, *n* = 48	UCS ^a^, *n* = 240	
Clavien-Dindo 30 days	288				0.354
1		126 (43.8%)	19 (39.6%)	107 (44.6%)	
2		70 (24.3%)	19 (20.8%)	60 (25.0%)	
3a		37 (12.9%)	8 (16.7%)	29 (12.1%)	
3b		32 (11.1%)	9 (18.8%)	23 (9.6%)	
4		6 (2.1%)	0 (0.0%)	6 (2.5%)	
5		8 (2.8%)	0 (0.0%)	8 (3.3%)	
Clavien-Dindo 90 days	280				0.176
1		46 (16.4%)	4 (8.3%)	42 (18.1%)	
2		71 (25.4%)	11 (22.9%)	60 (25.9%)	
3a		29 (10.4%)	8 (16.7%)	21 (9.0%)	
3b		4 (1.4%)	1 (2.1%)	3 (1.3%)	
4		1 (0.4%)	0 (0.0%)	1 (0.4%)	
5		11 (3.9%)	0 (0.0%)	11 (4.7%)	

^a^ *n* (%); ^b^ Fisher’s exact test.

**Table 4 cancers-17-00102-t004:** Factors determining the complications 30 days after RC.

Explanatory Variables	Severity of Complications (Clavien-Dindo)		
	B	CI 95%	*p*
Age years	0.01	−0.01–0.03	0.283
BMI kg/m^2^	−0.03	−0.06–−0.00	0.045
Fluids administrated (>1000 mL)	0.28	0.02–0.54	0.035
Fluids to surgery time to patient weight ratio ml/hr/kg	−0.04	−0.11–0.04	0.337
Neoadjuvant CHTx	−0.00	−0.30–0.30	0.999
Hypertension	0.11	−0.14–0.36	0.396
Diabetes mellitus	0.08	−0.19–0.35	0.572
Heart failure	−0.07	−0.37–0.24	0.669
Chronic kidney disease	0.11	−0.19–0.41	0.467
Hydronephrosis	0.09	−0.17–0.36	0.476
Acute kidney injury	0.11	−0.13–0.36	0.358
Urinary diversion (IC) *	-	-	-
Urinary diversion (UCS)	−0.53	−0.99–−0.07	0.024
aVBFI adjVBFI	−0.07 −0.27	−0.12–−0.03 −0.45–−0.08	0.002 0.005
Transfusion	0.24	−0.04–0.51	0.092
Hospitalization days	0.02	0.00–0.03	0.011
Reoperation	2.11	1.78–2.44	<0.001
aVBFI × Urinary diversion (UCS) adjVBFI × Urinary diversion (UCS)	0.07 0.24	0.02–0.12 0.06–0.43	0.011 0.011

Annotations: B—regression model coefficient; CI 95%—confidence interval 95%; *p*—*p*-value of the statistical test. *—No value is given for the urinary diversion (IC) variable because urinary diversions were compared to each other.

## Data Availability

Data is contained within the article.

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
