# Peer review of "Does the Administration of Intravenous Fluid Matter in the Context of the Incidence of Postoperative Complications After Radical Cystectomy?"

_cancers, 2024, doi:10.3390/cancers17010102_

Round 1
Reviewer 1 Report
Comments and Suggestions for Authors
The authors aimed to investigate the effects of intravenous fluid administration on postoperative complications in patients undergoing radical cystectomy (RC). The topic is not novel. Indeed, intravenous fluid management within the Enhanced Recovery After Surgery (ERAS) protocol, is safely adopted and recommended by guidelines. However, the sample size and the surgery performed (laparoscopic RC) are worthy of consideration. The study relied on 288 patients enrolled over a period of 5 years. Despite the efforts, several major issues should be addressed:
- Why did the authors expect fluid impact to increase the risk of complications? If the ERAS protocol has been followed and even more if the fluid administration is reasonable, there is no reason to be self-defeating.
- Any specific data on the neoadjuvant chemotherapy received? Several novel topics addressed the improved survival of RC patients underwent NAC and should be commented. Moreover, any data on the adjuvant chemo? (PMID 39424431, 38972785)
- Due to the discrepancies between the two groups a propensity score match between ureterocutaneostomies (UCS) and ileal conduit should be applied.
- The methodology is weak. The correlation does not explain causality. A multivariable logistic regression models should be completed to predict complications rate and severity. Moreover, an interaction analysis should also be performed for all the variables that may increase the risk of complications (such as ileal conduit and CKD or fluid administered and ileal conduit diversion). There is no sense in adding to the model "Acute kidney injury, transfusions, and hospitalization in days" that may be an effect of the complications or complications themselves.
- Was the lymph node dissection performed (PMID 39350629 , 39589370)? It may increase the complications rate.
- Any data on the infections? A recent paper analyzed the role of Alexis in avoiding those complications in RC patients (PMID 39443252 , 37940079);
- Any data on the paraplegic status? Does heart failure mean "heart valve history patients?
- Any data on CD complications more than 3? A specific paragraph should be added for those who may be relevant in the clinical management of RC patients
Author Response
Dear Reviewers,
The authors of the manuscript would like to thank for all critical comments and suggestions that are useful for improving the current draft. Please find below the explanation to Reviewer queries. We strongly believe that revised version of our draft will be suitable for publication in Cancers.
Kind regards, Pawel Lipowski & Kajetan Juszczak

Reviewer 2 Report
Comments and Suggestions for Authors
All in all a good and important paper.
Fasttrack and peritrack concepts are increasing in abdominal and pelvic surgeries wit good results.
The fluid intake should be restricted to max. 3L on the day of surgery according tot he guidelines.
Schwenk, W., Flemming, S., Girona-Johannkämper, M. et al. Strukturierte Fast-track-Implementierung zur beschleunigten Genesung nach elektiven kolorektalen Resektionen. Chirurgie 95, 148–156 (2024). https://doi.org/10.1007/s00104-023-01986-8
Please explain abbreviations when first used.
For older and comorbid patients, the uretero-ureterocutaneous fistula (UCN) without the use of intestinal interposition devices is an alternative and safe procedure. In our own patient population (n = 417), the operation time (incision-suture time: median 82 min; range 63-121 min) and mortality (2.1 %) are low despite a pronounced negative selection of comorbid, elderly patients (mean age 79 years). A potential disadvantage compared to the ileum conduit is the obligatory ureteral splinting.
Please add:
Dtsch Arztebl Int 2021; 118: 544; DOI: 10.3238/arztebl.m2021.0228
Author Response

(The authors gave the same response as above.)

Round 2
Reviewer 1 Report
Comments and Suggestions for Authors
No more comments.